# Dietary Ruminant and Industrial Trans-Fatty Acids Intake and Colorectal Cancer Risk

**DOI:** 10.3390/nu14224912

**Published:** 2022-11-20

**Authors:** Monireh Sadat Seyyedsalehi, Giulia Collatuzzo, Hamideh Rashidian, Maryam Hadji, Mahin Gholipour, Elham Mohebbi, Farin Kamangar, Eero Pukkala, Inge Huybrechts, Marc J. Gunter, Veronique Chajes, Paolo Boffetta, Kazem Zendehdel

**Affiliations:** 1Department of Medical and Surgical Sciences, University of Bologna, 40138 Bologna, Italy; 2Cancer Research Center, Cancer Institute, Tehran University of Medical Sciences, Tehran 1419733141, Iran; 3Health Sciences Unit, Faculty of Social Sciences, Tampere University, 33014 Tampere, Finland; 4Metabolic Disorders Research Center, Golestan University of Medical Sciences, Gorgan 4934174515, Iran; 5Department of Biology, School of Computer, Mathematical, and Natural Sciences, Morgan State University, Baltimore, MD 21251, USA; 6Finnish Cancer Registry-Institute for Statistical and Epidemiological Cancer Research, 00130 Helsinki, Finland; 7International Agency for Research on Cancer, 69372 Lyon, France; 8Stony Brook Cancer Center, Stony Brook University, Stony Brook, NY 11794, USA; 9Cancer Biology Research Center, Cancer Institute, Tehran University of Medical Sciences, Tehran 1419733141, Iran

**Keywords:** trans fatty acid, colorectal cancer, animal products, industrial fat, diet, elaidic acid, conjugated linoleic acid

## Abstract

As colorectal cancer (CRC) is largely due to modifiable lifestyle habits, the awareness on its risk factors is highly important. Dietary fatty acids have been linked to CRC risk. We explored the association between dietary trans fatty acids (TFAs) intake and CRC risk. We analyzed 865 CRC cases (434 in colon and 404 in rectum) and 3206 controls of the IROPICAN study, with data collected by trained interviewers using validated questionnaires. TFAs intake (industrial and ruminant types) was categorized into quartiles. Multivariate logistic regression models were built to calculate the odds ratios (OR) for the association between CRC and TFAs. We observed a positive association between industrial TFAs and colon cancer (OR for highest vs lowest quartile [OR_Q4vsQ1_] = 1.28, 95% confidence interval 1.07–1.54). A higher association was observed between industrial TFAs and CRC, occurring after 50 years of age. In addition, elaidic acid was associated with an increased risk of colon (OR_Q4vsQ1_ = 1.58, 1.24–2.02) and specifically of proximal colon cancer (OR _Q4vsQ1_ = 2.12, 1.40–3.20), as well as of rectum cancer (OR_Q4vsQ1_ = 1.40, 1.07–1.83). An inverse association was observed between ruminant TFAs intake and colon cancer risk (OR_Q4vsQ1_ = 0.80, 0.67–0.97). Industrial TFAs, such as semisolid/solid hydrogenated oils, may increase the risk of CRC, especially colon and proximal colon cancer. In contrast, ruminant TFAs do not appear to be associated with CRC. Awareness programs and regulatory actions regarding hydrogenated oils are warranted, given their high consumption through ultra-processed foods in more developed and less developed countries.

## 1. Introduction

Colorectal cancer (CRC) is the third and second most common cancer diagnosed in men (the rate standardized for age to the World Standard Population [ASR]: 23.4/100,000) and women (ASR: 16.2/100,000) around the world, respectively [1]. The incidence of CRC in Iran, a high-middle income country, has steadily increased from ASR 12.7/100,000 in 2016 to 19.9/100,000 in 2020 for both sexes, and the number of incident cases is predicted to increase by 54.1% from 11,558 in 2016 to 17,812 in 2025 [2]. Possible explanations for this trend can be found in (1) improved data collection methods related to the cancer registration system and cancer screening, which detect more new cases, as well as (2) changes in risk factors, including increased fat consumption, high body mass index (BMI), reduced physical activity, and increased smoking [3,4,5].

Regarding the first explanation, the first official cancer registry data were published in 1956 and publication continues until now [6]. Ministry of health launched a national NCD prevention program in Iran among population over 30 years to control cardiovascular, diabetes, lung diseases, and cancer among Iranian population. Screening of breast, cervical and CRC among average risk groups were included in this initiative. For CRC screening, fecal immunochemical test (FIT) among men and women 50–69 years old was recommended. There is no report available about the progress of this initiative as yet. However, a local report has been published from Isfahan province in the central part of Iran showed that 345,959 (41.5%) from the target population participated in the CRC screening in 2017–2021. Authors have reported a prevalence of 1593 (4.6 per 1000) for polyps and 132 (38.15 per 100,000) for cancer [7,8]. About second possible factor, several epidemiological studies found that types of fatty acids (FAs) are more strongly associated with a person’s health and disease incidence than total fat intake [5,9,10,11].

Trans fatty acids (TFAs) are unsaturated FAs that have one or more double bonds, found in ultra-processed human foods in two forms of industrial trans fatty acids (iTFAs) and ruminant trans fatty acids (rTFAs) [12,13]. In general, they are generated by chemical processes of industrial hydrogenation of vegetable (PHVO-TFA) and marine oils (PHFO-TFA), resulting in semi-solid or solid products at room temperature, which greatly extend storage and facilitates transportation [9,12,14]. This group includes several products, such as solid oils for daily cooking and high-fat industry-processed products (cakes and biscuits, margarine, snack foods) [15,16]. Moreover, animal products (e.g., beef, lamb, and dairy) are a minor source group of TFAs generated by rumen bacteria during the biohydrogenation of dietary unsaturated fats [9,14,17].

Some epidemiological studies raise the hypothesis that there is a positive association between chronic disease, particularly cancers and iTFA isomers, mainly (18:1n-9t) elaidic acid (EA). In addition, there are limited reliable data on the associations between rTFAs consumption and cancer risk, mostly (18:1n-7t) trans vaccenic acid (VA), (18:2 c9, t11 and 18:2 t10, c12) conjugated linoleic acids (CLAs), and (16:1n-9 t) palmitelaidic acid. The Iranian population experiences a high intake of TFAs, yet this risk factor has been poorly investigated in Iran [17,18,19,20]. Further research is needed to clarify the health effects and the carcinogenic potential of TFAs. Our investigation aimed at exploring the association between dietary iTFAs and rTFAs intake with the risk of CRC and its subsites in the large multicenter case-control study in Iran.

## 2. Methods

### 2.1. Study Design and Population

As part of the multicenter case-control study (IROPICAN), 898 CRC patients and 3233 healthy hospital visitor controls were recruited from the referral cancer hospitals seven provinces (Tehran, Fars, Mazandaran, Kerman, Golestan, Kermanshah, Khorasan-Razavi) between May 2017 and July 2020 [21]. Eligible cases were CRC incidence patients with pathological confirmation by an expert pathologist reviewer and defined according to the International Classification of Diseases (ICD-O-3) as tumors in the colon (C18), proximal colon (from cecum to the splenic flexure, ICD-O codes: C18.0 through C18.5), distal colon (from descending colon to sigmoid colon, ICD-O codes: C18.6 through C18.7), and rectum (from the recto-sigmoid junction (C19) down to the rectum). Tumors located at the border between two subsites of the colon (e.g., proximal and distal colon) which were involved to the same extent, the tumor was classified as an “overlapping lesions”. Additionally, all histological types of CRC except melanoma and sarcoma were included. Controls were enrolled concurrently with the cases among the relatives or friends of patients from non-oncology wards or those who visited the hospital for reasons other than receiving treatment. The controls had to be free of cancer at the date of recruitment. Participants who did not complete the food frequency questionnaire (FFQ) or those in the highest or lowest 1% distribution for the ratio of energy intake to estimated energy requirement were excluded (*n* = 60). Finally, this analysis included 865 CRC cases and 3206 healthy hospital visitor controls. The mean age at recruitment of the cases was 58.5 years, and of the controls 57.1.

### 2.2. Data Collection

Information on lifestyle and dietary intakes, education, tobacco smoking, opium use, socioeconomic status (SES), physical activity, medical history, and use of nonsteroidal anti-inflammatory drugs (NSAIDs) was obtained through face-to-face interviews by trained interviewers. At the time of enrollment, trained professionals measured the standing height of participants. Cases’ weights were drawn from the routine medical report or when the patients were able to stand up, we weighed them at the time of interview. Controls were weighed at the time of the interview. The BMI (weight/height squared, kg/m^2^) was calculated and classified according to the Centers for Disease Control and Prevention (CDC) data. Based on the Finnish Job Exposure Matrix (FINJEM) [22,23], we analyzed the estimated perceived physical activity workload (PPWL). A principal component analysis was used to define the socioeconomic status (SES) of the participants, based on the number of years of education the participants had and whether they owned any assets, such as vacuum cleaners, clothes washers, dishwashers, freezers, internet access, microwaves, laptops, mobile phones, cars, and shops [24].

### 2.3. Diet and Trans Fatty Acids Considered in the Analysis

In this study for estimation of TFAs, we considered (1) total TFAs; (2) total industrial TFAs; (3) (18:1n-9t) EA, (4) total ruminant TFAs, and (5) (18:2 c9, t11 and 18:2 t10, c12) CLAs. For this purpose, first, the intake of 131 food items was collected using the validated qualitative Persian Cohort food frequency questionnaire (FFQ) [25]. Then, we calculated the macro and micronutrient intakes specially different kind of FAs using the food composition database developed for the Iranian population, based on the USDA food composition [26], Near-East food composition [27], and Bahrain food composition tables [28]. Further information on the FFQ and calculated nutrients can be found elsewhere [29].

### 2.4. Statistical Analyses

All statistical analyses were carried out using Stata 14 (Stata Statistical Software: Release 14. College Station, TX, USA: Stata Corp LLC). We considered *p*-values < 0.05 as statistically significant.

According to the distribution of baseline characteristics and dietary intakes of the participants, categorical parameters were calculated as frequencies, and continuous variables were calculated as means and standard deviations (±SD). After examining the data distribution, all nutrient intakes were log-transformed to improve normality. We used the residual method to adjust dietary fatty acid intakes for total energy intake by regressing nutrient intakes on total energy intake derived from the FFQs. Multivariable-adjusted unconditional logistic regression models were used to compute odds ratios (ORs) and 95% confidence intervals (CI) of the association between dietary TFAs intakes and risk for CRC and its different subsites. All ORs were adjusted by gender (male, female), age (continuous), BMI (continuous), tobacco smoking, including cigarette smoking and water-pipe smoking (never, ever), opium use (never user, regular user, and non-regular user), province, SES (low, medium, and high), aspirin intake(Yes, No), physical activity (light, moderate, and heavy), processed meat intake including mortadella, hamburger and sausage (continuous, g/day), calcium (continuous, mg/day), fiber intake (continuous, g/day), and energy (continuous, Kcal/day). Quartiles were based on the distribution of different types of TFAs intake between controls. Continuous analyses were also run across quartiles in order to check the linear statistical trend. Additionally, we did stratified analyses according to gender and age (<50, ≥50 years) according to tertiles of TFAs intakes. In stratification models, *p*-value heterogeneity was assessed using a likelihood analysis (LR test).

## 3. Results

Out of the 865 CRC cases, 434 were from colon (145 from the proximal colon, 185 from the distal colon, and 104 were overlapping lesions); 404 from rectum and 27 cases from unknown CRC subsite. Two thirds of the CRC cases were male. Approximately 76% of all CRC cases (671) were older than 50 at the diagnosis date. More than 50% of the CRC cases were from Fars and Tehran. Table 1 shows the socio-demographic characteristics, dietary factors intakes, and the distribution of selected risk factors.

We observed positive associations of total TFAs, iTFAs, and EA intake with CRC risk, with a relation between total TFAs and CRC [OR comparing highest to the lowest quartile, [OR_Q4vsQ1_ = 1.10, 95% CI 1.02–1.19], [OR_Q4vsQ1_ = 1.17, 95% CI 1.02–1.34], and [OR_Q4vsQ1_ = 1.43, 95% CI 1.19–1.73] respectively (Table 2). Subsite analyses showed positive associations between iTFAs and colon cancer [OR_Q4vsQ1_ = 1.28, 95% CI 1.07–1.54], and EA and colon [OR_Q4vsQ1_ = 1.58, 95% CI 1.24–2.02], proximal colon [OR_Q4vsQ1_ = 2.12, 95% CI 1.40–3.20], and rectum cancer [OR_Q4vsQ1_ = 1.40, 95% CI 1.07–1.83]. An inverse relationship was observed between rTFAs and CRC [OR_Q4vsQ1_ = 0.85, 95% CI 0.73–0.98], and colon cancer [OR_Q4vsQ1_ = 0.80, 95% CI 0.67–0.97].

Analyses stratified by age group showed that the association between iTFAs (P-heterogeneity = 0.02) and CRC was significant among subjects older than 50. Males also had a high positive association between iTFAs intake and colon cancer (P-heterogeneity = 0.03). Conversely, rTFAs intakes had a stronger inverse association with CRC in females (P-heterogeneity = 0.02) and participants older than 50 years (P-heterogeneity = 0.04). The association between other anatomical sites and the different types of TFAs was not statistically significant (Table 3).

## 4. Discussion

We found that high levels of TFAs are associated with the risk of CRC. Although a high intake of iTFAs was associated with an increased risk of CRC, rTFAs had the opposite effect. Previous studies have reported inconsistent results for the association between TFAs consumption and CRC risk [17,30,31,32,33,34]. Two cohort studies conducted on women found no association between TFAs consumption and overall risk of CRC or different subsites [30,33]. However, two case-control studies indicated an association between TFAs consumption and colon cancer [32] and CRC risk [35] among women but not men. An additional case-control study found that higher consumption of TFAs is associated with a higher risk of rectum and distal colon cancers in White subjects, while no association was found among African Americans [34]. A nested case-control study reported that dietary iTFAs, especially EA, was positively associated with rectal cancer risk [5]. In vivo studies showed that EA may significantly increase cell growth, anti-apoptotic survival, and invasion in different CRC cell types in a dose-dependent manner, demonstrating its pro-metastatic potential [36]. Furthermore, previous evidence suggests that TFAs may decrease in plasma HDL and increase LDL concentrations and exert pro-inflammatory and pro-arrhythmogenic actions [37]. Moreover, a study that estimated the intake of TFAs based on consumption of foods containing partially hydrogenated vegetable oils established a significant association between oils and condiments and CRC [38]. According to the review by Thompson and coauthors, evidence is insufficient to support an adverse effect of TFAs on CRC [39].

A possible explanation for the inconsistency in the results regarding the association between TFAs and CRC may be the heterogeneity of these micronutrients. When considering total TFAs, we observed a positive association with CRC, but when separating iTFAs from rTFAs we could identify a direct relationship for the iTFAs and an inverse relationship for rTFAs. Moreover, when stratifying by anatomical subsite, these associations were particularly evident for proximal colon, where EA increased the risk, but CLAs decreased the risk of cancer. To our knowledge, this is the first study to investigate the association of both iTFAs and rTFAs by anatomical subsite of CRC.

The association between rTFA intake and cancer risk has not been deeply investigated in humans, but several in vivo and animal studies found reverse associations between some types of rTFAs (including VA, and CLAs) and CRC [7,40]. A systematic review on the impact of CLAs on body composition and energetic metabolism suggested that eating foods naturally enriched with CLAs during a lifetime might reduce adiposity [41] and the concomitant risks associated with obesity, such as CRC [42]. Previous studies on ghee (clarified butter), a fat-rich dairy product and a source of rTFAs commonly eaten in Iran, did not report an association with CRC risk; however, some reports suggested its beneficial effect on human health [40,41,43]. Since ghee contains a combination of fatty acids (more than 50% saturated fat, 20% monounsaturated fatty acids, 1 to 3% polyunsaturated fatty acids, and 3.5% rTFA) [43,44], further research is needed to clarify its effect on CRC. Studies on TFAs intakes and CRC risk in low- and middle-income countries, specifically in the Eastern Mediterranean Region (EMRO), are limited. On the other hand, according to a WHO report, the EMRO region has one of the highest levels of TFAs in its food supply [45,46]. The systematic review examined 29 countries between 1995 and 2017. As a result, TFAs consumption is still higher than WHO-recommended levels (less than 1% of total energy intake, or less than 2.2 g/day, for a 2000 calorie/day diet) in several countries, including both more developed countries (e.g., Canada and the United States) as well as less developed countries (e.g., Iran and Lebanon) [47,48]. A WHO report showed that Iranian households consume large amounts of cooking oil from partially hydrogenated vegetable oils, a major source of iTFAs, especially EA; iTFAs accounted for more than 12% of the total daily calories intake [19]. In addition, some studies point to the high intake of solid oils and Western dietary products with high content of TFAs as risk factors of CRC in different population, such as the Iranians [49].

Many interventions have been carried out in recent years to increase awareness and reduce the consumption of solid oils. However, these products are still used in various food industries and in household cooking [50]. It is thought that the change in solid oils consumption and iTFAs intake during the last twenty years [19,51,52] may impact the incidence of CRC.

Our study has several strengths. The possibility to distinguish the effect of iTFAs and rTFAs is one of the main strengths. To our knowledge, this is the first study to assess the effect of the consumption of total and different types of iTFAs and rTFAs on CRC and its subsites. In the present study, lifestyle, and dietary intake data, as well as information on a wide range of potential CRC risk factors, were collected by a trained interviewer in the same interview setting with the use of similar food photographs and other questioning tools in all centers under the supervision of trained observers. The detailed information allowed for adjusting our analysis by several confounders. Moreover, we only included cases with pathological confirmation and expert review.

Our study has some weaknesses. Using FFQ for data collection is prone to recall bias because it is a retrospective method and dependent on the participant’s memory. Furthermore, some people have misreported their consumption of certain foods according to their personal characteristics, such as obesity, gender, age and so on. Furthermore, other errors in the assessment of diet can be introduced through the use of FCTs to calculate different nutrient intakes. This can be due to natural variations or a lack of knowledge regarding the composition of processed and packaged foods as well as foods prepared outside of home. Moreover, we used healthy visitors without cancer as controls rather than an unselected population [53].

## 5. Conclusions

According to these results, iTFAs, such as those found in semisolid/solid hydrogenates oils and in snack foods that use this type of oil in their production, may be implicated in CRC development. In contrast, rTFAs, specifically CLAs, appeared to be associated with lower CRC risk. Following the WHO report indicating a high consumption of hydrogenated oils in both more developed and less developed countries, it is essential to organize the pieces of training and awareness programs as well as regulatory actions regarding the hazards of such products. Our results highlight the need to pay more attention to nutrition control in Iranians and among populations with similar cultural and geographical characteristics to Iran, particularly those of the EMRO region.

## Figures and Tables

**Table 1 nutrients-14-04912-t001:** Selected baseline demographic and lifestyle characteristics of study participants by colorectal cancer status, IROPICAN study.

	Controls	Cases		
		Colorectal *	Colon	Rectum
**Total, N (%)**	**3206 (100%)**	**865 (100%)**	**434 (100%)**	**404 (100%)**
**Province, N (%)**				
Tehran	816 (25%)	165 (19%)	101 (23%)	64 (15%)
Fars	943 (29%)	248 (28%)	93 (21%)	155 (38%)
Kerman	525 (16%)	100 (11%)	49 (11%)	51 (12%)
Golestan	373 (11%)	149 (17%)	89 (20%)	53 (13%)
Mazandaran	136 (4%)	59 (6%)	34 (7%)	25 (6%)
Kermanshah	251 (7%)	68 (7%)	31 (7%)	35 (8%)
Mashhad	162 (5%)	76 (8%)	37 (8%)	21 (5%)
**Gender, N (%)**				
Women	1003 (31.28%)	368 (42.54%)	193 (44.47%)	169 (41.83%)
Men	2203 (68.71%)	497 (57.46%)	241 (55.53%)	235 (58.17%)
**Age at interview, years, N (%)**				
<30	21 (0.66%)	8 (0.92%)	3 (0.69%)	5 (1.24%)
>=30 & <40	227 (7.08%)	60 (6.94%)	32 (7.37%)	27 (6.68%)
>=40 & <50	503 (15.69%)	126 (14.57%)	64 (14.75%)	58 (14.36%)
>=50 & <60	993 (30.97%)	242 (27.98%)	112 (25.81%)	123 (30.45%)
>=60 & <70	1020 (31.82%)	258 (29.83%)	137 (31.57%)	112 (27.72%)
>=70	442 (13.79%)	171 (19.77%)	86 (19.82%)	79 (19.55%)
**SES, N (%)**				
Low	861 (26.86%)	337 (38.27%)	159 (36.64%)	161 (39.85%)
Moderate	1078 (33.62%)	234 (27.05%)	118 (27.19%)	109 (26.98%)
High	1267 (39.52%)	300 (34.68%)	157 (36.18%)	134 (33.17%)
**Tobacco consumption, N (%)**				
Never	2153 (67.16%)	629 (72.72%)	334 (76.96%)	274 (67.82%)
Ever	1053 (32.84%)	236 (27.28%)	100 (23.04%)	130 (32.18%)
**Opium consumption, N (%)**				
Never use	2646 (82.53%)	731 (84.51%)	369 (85.02%)	340 (84.16%)
Regular user	432 (13.47%)	88 (10.17%)	40 (9.22%)	46 (11.39%)
Non-regular user	128 (3.99%)	46 (5.32%)	25 (5.76%)	18 (4.46%)
**Physical activity workload, N (%)**				
Sedentary	1034 (32.27%)	287 (33.18%)	147 (33.87%)	132 (32.67%)
Moderate	701 (21.88%)	155 (17.92%)	78 (17.97%)	72 (17.82%)
Heavy	694 (21.66%)	184 (21.27%)	87 (20.05%)	87 (21.53%)
Unknown	775 (24.19%)	239 (27.63%)	122 (28.11%)	113 (27.97%)
**Aspirin use, N (%)**				
No	2469 (77%)	709 (81.97%)	358 (82.49%)	327 (80.94%)
Yes	737 (22.99%)	156 (18.03%)	76 (17.51%)	77 (19.06%)
**BMI, kg/m^2^, mean (±SD)**	26.6 (±4.72)	26.9 (±4.99)	26.9 (±5.07)	26.8 (±4.85)
**BMI, N (%)**				
Underweight (<18.5)	90 (2.81%)	28 (3.24%)	14 (3.23%)	14(3.47%)
Normoweight (>=18.5 & <25)	1121 (34.97%)	261 (30.17%)	135 (31.11%)	119(29.46%)
Overweight (>=25 & <30)	1311 (40.89%)	371 (42.89%)	184 (42.40%)	177 (43.81%)
Obese (>=30)	684 (21.33%)	205 (23.70%)	101 (23.27%)	94(23.27%)
**Dietary intake, mean(±SD)**				
Total processed meat (g/day)	1.9 (±0.12)	2.2 (±0.26)	2.5 (±0.43)	1.8 (±0.31)
Fiber (g/day)	24.7 (±11)	26.8 (±12)	25.2 (±12)	26.3 (±13)
Calcium (mg/day)	860.3 (±6.6)	908.2 (±14.7)	920.6 (±20.9)	880.2 (±21.2)
Dietary energy intake (Kcal/day)	2319.4 (±878)	2405.6 (±1076)	2387.2 (±1082)	2393.2 (±1066)
Dietary energy intake from total fat (Kcal/day)	616.68 (±261)	696.51 (±351)	714.6 (±351)	667.71 (±351)
Dietary energy intake from total TFAs (Kcal/day)	3.78 (±2.52)	4.14 (±2.97)	4.41 (±3.06)	3.78 (±2.79)
**Trans fatty acids (TFA) Dietary intakes(g/day), mean (±SD)**				
Total fat	68.52 (±29.92)	77.39 (±39.70)	79.40 (±39.71)	74.19 (±39.29)
Total trans fatty acid	0.42 (±0.28)	0.46 (±0.33)	0.49 (±0.34)	0.42 (±0.31)
Total industrial trans fatty acids	0.32 (±0.21)	0.36 (±0.26)	0.38 (±0.26)	0.32 (±0.24)
EA	0.30 (±0.22)	0.34 (±0.26)	0.36 (±0.26)	0.31 (±0.25)
Total trans ruminant fatty acids	0.10 (±0.07)	0.11 (±0.08)	0.11 (±0.09)	0.10 (±0.07)
CLAs	0.05 (±0.04)	0.06 (±0.05)	0.06 (±0.05)	0.05 (±0.04)

* Includes 27 cases with unknown subsite. Abbreviations: BMI, body mass index; SES, socio-economic status; CLAs, conjugated linoleic acids; EA, elaidic acid.

**Table 2 nutrients-14-04912-t002:** Odds ratios and 95% confidence intervals of the association between dietary intake of trans fatty acids and colorectal cancer risk, by anatomical subsite.

Trans Fatty Acids		Colorectal Cancer			Colon			Proximal Colon			Distal Colon			Rectum		
		Mean (g/day)	Num. Case	Multivariable OR (95%CI)	Mean (g/day)	Num. Case	Multivariable OR (95%CI)	Mean (g/day)	Num. Case	Multivariable OR (95 %CI)	Mean (g/day)	Num. Case	Multivariable OR (95 %CI)	Mean (g/day)	Num. Case	Multivariable OR (95 %CI)
**Total trans fatty acid**	Q1	0.267	219	Ref.	0.293	108	Ref.	0.258	30	Ref.	0.292	52	Ref.	0.234	105	Ref.
	Q2	0.374	196	1.13 (0.89–1.45)	0.388	100	1.09 (0.79–1.51)	0.411	33	0.87 (0.50–1.52)	0.336	42	1.07 (0.67–1.72)	0.351	89	1.22 (0.87–1.70)
	Q3	0.481	201	**1.38 (1.07–1.77)**	0.506	90	**1.42 (1.02–1.98)**	0.498	34	1.051 (0.60–1.84)	0.512	32	1.31 (0.81–2.13)	0.454	106	1.36 (0.96–1.93)
	Q4	0.695	249	**1.31 (1.02–1.68)**	0.723	136	1.29 (0.93–1.80)	0.698	48	1.53 (0.91–2.58)	0.715	59	1.15 (0.71–1.88)	0.634	104	1.37 (0.97–1.94)
	Q4 vs. Q1			**1.10 (1.01–1.19)**			1.10 (0.99–1.22)			1.17 (0.99–1.39)			1.06 (0.91–1.24)			1.10 (0.99–1.23)
	p-trend			**0.015**			0.061			0.064			0.397			0.064
**Total industrial trans fatty acids**	Q1	0.116	222	Ref.	0.115	92	Ref.	0.123	29	Ref.	0.105	43	Ref.	0.117	126	Ref.
	Q2	0.240	195	**1.36 (1.01–1.84)**	0.240	95	**1.63 (1.08–2.46)**	0.242	35	**2.34 (1.18–4.63)**	0.238	41	0.98 (0.54–1.76)	0.241	96	1.15 (0.78–1.71)
	Q3	0.332	180	**1.53 (1.05–2.22)**	0.334	93	1.91 (1.15–3.16)	0.333	27	2.148 (0.93–4.95)	0.333	43	1.72 (0.85–3.47)	0.329	81	1.32 (0.80–2.19)
	Q4	0.613	265	**1.73 (1.12–2.68)**	0.652	154	**2.47 (1.40–4.37)**	0.638	54	**2.94 (1.15–7.51)**	0.670	58	**2.32 (1.02–5.28)**	0.661	101	1.23 (0.67–2.27)
	Q4 vs. Q1			**1.17 (1.02–1.34)**			**1.28(1.07–1.54)**			1.32 (0.98–1.77)			**1.33 (1.02–1.73)**			1.07 (0.88–1.31)
	p-trend			**0.026**			**0.006**			0.064			**0.033**			0.468
**Elaidic acid**	Q1	0.099	224	Ref.	0.102	100	Ref.	0.106	33	Ref.	0.098	48	Ref.	0.098	122	Ref.
	Q2	0.224	198	**1.68 (1.18–2.41)**	0.227	88	**2.07 (1.28–3.33)**	0.230	33	**5.46 (2.41–12.42)**	0.230	38	0.78 (0.38–1.62)	0.222	103	1.45 (0.88–2.39)
	Q3	0.317	171	**2.52 (1.56–4.07)**	0.319	90	**3.47 (1.85–6.50)**	0.325	26	**10.22 (3.37–30.96)**	0.313	38	1.43 (0.56–3.62)	0.311	77	**2.49 (1.25–4.95)**
	Q4	0.601	272	**2.69 (1.46–4.93)**	0.635	156	**3.98 (1.82–8.73)**	0.620	53	**13.78 (3.50–54.21)**	0.649	61	1.33 (0.41–4.35)	0.653	102	**2.45 (1.01–5.93)**
	Q4 vs. Q1			**1.43 (1.19–1.73)**			**1.58 (1.24–2.02)**			**2.12 (1.40–3.20)**			1.15 (0.80–1.66)			**1.40 (1.07–1.83)**
	p-trend			**0.000**			**0.000**			**0.000**			0.428			**0.012**
**Total trans ruminant fatty acids**	Q1	0.030	220	Ref.	0.031	101	Ref.	0.032	32	Ref.	0.031	49	Ref.	0.029	116	Ref.
	Q2	0.070	220	**0.63 (0.47–0.85)**	0.073	96	**0.52 (0.35–0.78)**	0.074	40	0.66 (0.35–1.26)	0.073	39	0.65 (0.36–1.15)	0.069	120	0.73 (0.49–1.07)
	Q3	0.107	161	**0.43 (0.29–0.64)**	0.106	89	**0.42 (0.25–0.69)**	0.106	22	0.54 (0.24–1.19)	0.106	42	**0.31 (0.14–0.67)**	0.107	66	**0.40 (0.23–0.68)**
	Q4	0.187	264	**0.60 (0.39–0.93)**	0.201	148	**0.46 (0.26–0.82)**	0.193	51	0.42 (0.16–1.08)	0.211	55	**0.35 (0.15–0.82)**	0.193	102	0.70 (0.38–1.28)
	Q4 vs. Q1			**0.85 (0.73–0.98)**			**0.80 (0.67–0.97)**			0.75 (0.55–1.02)			**0.71 (0.54–0.94)**			0.86 (0.70–1.05)
	p-trend			**0.031**			**0.026**			0.072			**0.017**			0.163
**Conjugated linoleic acid(CLAs)**	Q1	0.017	218	Ref.	0.017	99	Ref.	0.018	33	Ref.	0.017	46	Ref.	0.016	116	Ref.
	Q2	0.040	219	**0.55 (0.38–0.80)**	0.040	95	**0.42 (0.25–0.69)**	0.040	37	**0.36 (0.15–0.85)**	0.039	41	0.79 (0.36–1.69)	0.039	120	0.68 (0.41–1.15)
	Q3	0.059	164	**0.34 (0.19–0.60)**	0.059	90	**0.25 (0.12–0.53)**	0.059	25	**0.20 (0.06–0.72)**	0.059	42	**0.28 (0.08–0.89)**	0.060	69	**0.37 (0.16–0.84)**
	Q4	0.106	264	0.55 (0.28–1.09)	0.114	150	**0.32 (0.13–0.79)**	0.112	50	**0.18 (0.04–0.85)**	0.120	56	0.41 (0.10–1.66)	0.113	99	0.79 (0.29–2.13)
	Q4 vs. Q1			0.93 (0.75–1.15)			0.83 (0.63–1.11)			0.75 (0.46–1.20)			0.85 (0.55–1.30)			0.98 (0.72–1.34)
	p-trend			0.523			0.218			0.239			0.461			0.940

OR comparing highest to lowest quartile (Q4 vs. Q1). Ranges in bold face are statistically significant. Total industrial trans fatty acids included (18:1n-9t) elaidic acid, (18:2n-6tt)trans octadecadienoic acid. Total ruminant trans fatty acids included (16:1n-9t) palmitelaidic acid, (18:1n-7t) vaccenic acid, (18:2 c9, t11 and 18:2 t10, c12) conjugated linoleic acid. Adjusted by province, age, socioeconomic status, gender, aspirin intake, body mass index, tobacco, opium use, physical activity, processed meat, fiber intake, calcium, energy intake.

**Table 3 nutrients-14-04912-t003:** The multivariable odds ratios (OR) and 95% confidence intervals (CI) of colorectal cancer and its subsites according to tertiles of trans fatty acids intakes, stratified by gender and age. Q1: reference.

Anatomical Location	Trans Fatty Acids Type	Tertiles	Gender			Age		
			Male (N_case_ = 497/ N_control_ = 2203)	Female (N_case_ = 368/ N_control_ = 1003)	P_heterogeneity **	<= 50 (N_case_ = 206/ N_control_ = 871)	>50 (N_case_ =659/ N_control_ = 2335)	P_heterogeneity **
			OR (95% CI)	OR (95% CI)		OR (95% CI)	OR (95% CI)	
**Colorectal**	**Total**	**Q2**	**1.28 (0.97–1.69)**	**1.11 (0.79–1.56)**		1.54 (0.97–2.43)	1.18 (0.93–1.51)	
		Q3	1.22 (0.92–1.63)	1.33 (0.94–1.87)		1.12 (0.71–1.75)	**1.46 (1.13–1.88)**	
		Q3 vs. Q1*	1.10 (0.96–1.27)	1.16 (0.97–1.37)	0.21	1.03 (0.83–1.28)	**1.21 (1.06–1.37)**	0.12
	**Industrial**	Q2	1.07 (0.74–1.55)	1.01 (0.65–1.57)		0.90 (0.51–1.58)	1.14 (0.83–1.58)	
		Q3	1.28 (0.79–2.06)	1.66 (0.93–2.97)		0.82 (0.41–1.64)	**1.76 (1.15–2.70)**	
		Q3 vs. Q1	1.11 (0.88–1.41)	1.29 (0.96–1.71)	0.14	0.86 (0.61–1.21)	**1.31 (1.06–1.62)**	**0.02**
	**Ruminant**	Q2	0.86 (0.59- 1.24)	**0.61 (0.39–0.96)**		0.83 (0.47–1.47)	**0.69 (0.50–0.96)**	
		Q3	0.89 (0.55–1.44)	**0.54 (0.29–0.99)**		1.06 (0.53–2.12)	**0.63 (0.40–0.98)**	
		Q3 vs. Q1	0.95 (0.75–1.21)	0.74 (0.55–1.00)	**0.02**	1.08 (0.76–1.52)	**0.80 (0.64–0.99)**	**0.04**
**Colon**	**Total**	Q2	1.14 (0.78–1.67)	1.21 (0.78–1.88)		1.40 (.77–2.54)	1.17 (0.84–1.63)	
		Q3	1.22 (0.83–1.79)	**1.56 (1.01–2.42)**		0.92 (0.51–1.68)	**1.73 (1.24–2.42)**	
		Q3 vs. Q1	1.10 (0.91–1.33)	**1.25 (1.01–1.55)**	0.76	0.93 (0.70–1.23)	**1.32(1.12–1.56)**	0.13
	**Industrial**	Q2	1.53 (0.91–2.57)	0.68 (0.38–1.22)		0.76 (0.35–1.64)	1.24 (0.79–1.94)	
		Q3	**2.32 (1.22–4.39)**	1.01 (0.48–2.13)		0.77 (0.30–1.93)	**2.21 (1.26–3.89)**	
		Q3 vs. Q1	**1.48 (1.08–2.02)**	1.02 (0.70–1.47)	**0.03**	0.82 (0.52–1.27)	**1.48 (1.12–1.95)**	0.13
	**Ruminant**	Q2	0.70 (0.42–1.15)	0.98 (0.54–1.78)		1.01 (0.46–2.18)	0.67 (0.43–1.05)	
		Q3	0.60 (0.32–1.13)	0.93 (0.42–2.08)		1.45 (0.57–3.67)	**0.49 (0.27–0.89)**	
		Q3 vs. Q1	0.79 (0.58–1.08)	0.96 (0.65–1.41)	0.21	1.29 (0.82–2.02)	**0.71 (0.54–0.95)**	0.43
**Rectum**	**Total**	Q2	**1.65 (1.14–2.40)**	0.91 (0.57–1.48)		**1.92 (1.00–3.68)**	1.23 (0.89–1.71)	
		Q3	1.40 (0.93–2.09)	0.99 (0.62–1.60)		1.40 (0.74–2.67)	1.24 (0.87–1.75)	
		Q3 vs. Q1	1.18 (0.97–1.44)	1.01 (0.79–1.28)	0.12	1.14 (0.84–1.54	1.11 (0.93–1.32)	0.30
	**Industrial**	Q2	0.82 (0.50–1.35)	1.50 (0.83–2.69)		1.06 (0.49–2.28)	1.10 (0.72–1.70)	
		Q3	0.74 (0.38–1.46)	**2.75 (1.20–6.28)**		0.87 (0.33–2.33)	1.44 (0.79–2.60)	
		Q3 vs. Q1	0.86 (0.62–1.19)	**1.67 (1.11–2.49)**	0.30	0.91 (0.56–1.48)	1.19 (0.89–1.59)	0.17
	**Ruminant**	Q2	1.00 (0.61–1.64)	**0.37 (0.20–0.68)**		0.68 (0.32–1.46)	0.65 (0.42–1.02)	
		Q3	1.19 (0.61–2.34)	**0.32 (0.14–0.75)**		0.69 (0.26–1.82)	0.70 (0.38–1.29)	
		Q3 vs. Q1	1.08 (0.77–1.51)	**0.56 (0.37–0.84)**	0.07	0.85 (0.52–1.38)	0.83 (0.62–1.11)	0.06

* OR comparing highest to lowest tertiles (Q3 vs. Q1) ** *p*-value from likelihood ratio test Results in bold are statistically significant (*p* < 0.05). Total industrial trans fatty acids included (18:1n-9t) elaidic acid, (18:2n-6tt) trans octadecadienoic acid. Total ruminant trans fatty acids included (16:1n-9t) Palmitelaidic acid, (18:1n-7t) Vaccenic acid, (18:2 c9, t11 and 18:2 t10, c12) conjugated linoleic acid. Adjusted by province, age, socioeconomic status, gender, body mass index, aspirin intake, tobacco, opium use, physical activity, processed meat, fiber intake, calcium, energy intake.

## Data Availability

The data supporting this study’s findings are available from the corresponding author upon reasonable request.

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
