# Peer review of "Dietary Ruminant and Industrial Trans-Fatty Acids Intake and Colorectal Cancer Risk"

_nutrients, 2022, doi:10.3390/nu14224912_

Round 1

Reviewer 1 Report

This paper surveys trans-fat consumption in Iran against risk for colon cancer in sub-regions, distinguishing ruminant, and industrial sources.  The general finding is increased incidence of cancer in the higher quartiles of industrial consumption, where quartiles were established from a large control population.  Ruminant sources reduced incidence.  In general, the study seems fairly well done given the use of surveys and dietary Tables.  I find that the Discussion overstates some findings, for rectum, in particular; first line of Discussion..    The Abstract mentions analysis of Q4 vs. Q1 that is not found in the Tables, I had access to.  Please add this to Table 2 or create a new Table with this information or revise to conform with major analysis in Table 3 of Q3 vs Q1..

Reviewer 2 Report

Dear Authors,

I would like to congratulate you with this fine study.

I have some questions:

INTRODUCTION

--could you please elaborate more on CRC screening program (when was it started in IRAN, how is it performed, what is the participation rate). 

--line 62 "also" change to "moreover". 

METHODS

--you have collected the dietary habits of one year before being diagnosed with CRC. We do know that it takes at least 5-10 years for CRC to evolve. Can you elaborate more?

--what do you mean by "trained interviewer"?

--lines 107-114 - the reference numbering should be re-arranged

RESULTS

--what is the overlapping lesion??

--did you case matched the groups to all other risk factors?

--I cannot see the tables

DISCUSSION

--well written, no comments

Reviewer 3 Report

The manuscript presents the results of a study of the relationship between consumption of dietary iTFAs and rTFAs with the risk of colorectal cancer. The authors presented interesting and valuable results, but it is a pity that it is limited to one population despite multi-center authorship.

In my opinion, the introduction needs to be improved. Information should be added on the impact of total fat on CRC; on the amount of TFAs in products that may be the main sources of them in the diet. What other nutritional factors are associated with the higher risk CRC? Then fiber, calcium, meat appear in the Results.

In the Methods - add information what references were used to categorize BMI.  On the basis of which data the TFAs was estimated?

In the Results and table 1 - add proper interpretation of nutrients intake for group - assess the prevalence of in-/adequate intakes apart from means. What was the percentage of energy from total fat and TFAs in the diet?

Minor comments:

- remove headings from Abstract

- Table 2:  18:1n-7t (elaidic acid) and18:1n-7t (Vaccenic acid)?;18:2n-6tt - name? 

- the manuscpit requires editorial attention

Reviewer 4 Report

nutrients-1996277

Dietary ruminant and industrial trans-fatty acids intake and colorectal cancer risk

General Comments

This manuscript describes the relationship between TFA and incidence of CRC, using data from the IROPICAN study. Although I am not a statistics expert, the statistical analyses appear to be appropriate. English usage is quite good.

Specific comments

L. 61. What is meant by “daily solid oils”?

L. 68. According to the Merck Index (albeit a 1983 edition), vaccenic acid is indeed 18:1 n-7t. However, newer gas/liquid chromatographs now indicate significant amounts of 18:1 n-7c, so it may be more appropriate to indicate 18:1 7t as “trans-vaccenic acid”. Also, 18:1 t11 would be preferred to 18:1 7t, being more consistent with your descriptions of the CLAs such as 18:2 c9, t11.

L. 108-109. Why wasn’t 18:2 t10, c12 included in your analyses? Its biological effects differ greatly from 18:2 c9, t11.

L. 131 and Table 2 legend. In the U.S., hamburger typically pure ground beef, with none of the salts and nitrates commonly added to processed meats. The situation could be quite different in Iran but, if not, including hamburger to adjust ORs is inappropriate.

L. 142-143. Table 1 does not indicate statistical differences in BMI or physical activity between cases and controls.

Table 2. I assume that ORs and ranges in bold face are statistically significant. This should be indicated in the table legend.

L. 164. I believe this should be “P-heterogeneity=0.04”, not 0.4.

L. 184. Please change “Whites” to “White”.

L. 191-193. Please provide a reference/citation to support this statement.

L. 204-206. Please indicate that ruminal vaccenic acid is 18:2 c11, i.e., cis-vaccenic.

L. 232-234. Younger people probably consume less TFAs, but older individuals (especially men) are more susceptible to colon cancer just because they are older. I suggest deleting this sentence.

Round 2

Reviewer 1 Report

The authors have adequately addressed my concerns.